# Chagas disease is related to structural changes of the gut microbiota in adults with chronic infection (TRIPOBIOME Study)

José A. Pérez-Molina[1,2,3]*, Clara Crespillo-Andújar[1,2,3], Elena Trigo[3,4], Sandra Chamorro[1,2,3], Marta Arsuaga[3,4], Leticia Olavarrieta[5], Beatriz Navia[6,7], Oihane Martín[2,3,8], Begoña Monge-Maillo[1,2,3], Francesca F. Norman[1,2,3], Val F. Lanza[2,9], Sergio Serrano-Villar[2,3,10]

1 National Referral Centre for Tropical Diseases, Infectious Diseases Department, Hospital Universitario Ramón y Cajal, Madrid, Spain, 2 Instituto Ramón y Cajal de Investigación Sanitaria (IRYCIS), Madrid, Spain, 3 CIBER de Enfermedades infecciosas, Instituto de Salud Carlos III, Madrid, Spain, 4 Imported Diseases and International Health Referral Unit. High Level Isolation Unit. La Paz- Carlos III University Hospital, Madrid, Spain, 5 Translational Genomics Unit. Hospital Universitario Ramón y Cajal, Instituto Ramón y Cajal de Investigación Sanitaria (IRYCIS), Madrid, Spain, 6 Department of Nutrition and Food Science, Faculty of Pharmacy, Universidad Complutense de Madrid, Madrid, Spain, 7 Research Group VALORNUT-UCM (920030), Universidad Complutense de Madrid, Madrid, Spain, 8 Microbiology Department, Hospital Universitario Ramón y Cajal, Madrid, Spain, 9 Bioinformatics Unit, Hospital Universitario Ramón y Cajal, Madrid, Spain, 10 Infectious Diseases Department, Hospital Universitario Ramón y Cajal, Madrid, Spain

* jperezm@salud.madrid.org

## Abstract

### Background

The implications of the gut microbial communities in the immune response against parasites and gut motility could explain the differences in clinical manifestations and treatment responses found in patients with chronic Chagas disease.

### Methodology/Principal findings

In this pilot prospective cross-sectional study, we included 80 participants: 29 with indeterminate CD (ICD), 16 with cardiac CD (CCD), 15 with digestive CD (DCD), and 20 controls without CD. Stool was collected at the baseline visit and faecal microbial community structure DNA was analyzed by whole genome sequencing. We also performed a comprehensive dietary analysis. Ninety per cent (72/80) of subjects were of Bolivian origin with a median age of 47 years (IQR 39–54) and 48.3% (29/60) had received benznidazole treatment. There were no substantial differences in dietary habits between patients with CD and controls. We identified that the presence or absence of CD explained 5% of the observed microbiota variability. Subjects with CD exhibited consistent enrichment of *Parabacteroides spp*, while for *Enterococcus hirae*, *Lactobacillus buchneri* and *Megamonas spp*, the effect was less clear once excluded the outliers values. Sex, type of visceral involvement and previous treatment with benznidazole did not appear to have a confounding effect on gut microbiota structure. We also found that patients with DCD showed consistent *Prevotella spp* enrichment.

**Data Availability Statement:** All relevant data are within the manuscript and its Supporting Information files.

**Funding:** This research was supported by CIBER -Consorcio Centro de Investigación Biomédica en Red- (CB 2021), Instituto de Salud Carlos III, Ministerio de Ciencia e Innovación and Unión Europea – NextGenerationEU to JAPM. The funders had no role in study design, data collection and analysis, decision to publish, or preparation of the manuscript.

**Competing interests:** The authors have declared that no competing interests exist.

## Conclusions

We found a detectable effect of Chagas disease on overall microbiota structure with several potential disease biomarkers, which warrants further research in this field. The analysis of bacterial diversity could prove to be a viable target to improve the prognosis of this prevalent and neglected disease.

## Author summary

Chagas disease (CD) affects about 6 million people in endemic areas of the Americas and more than 500,000 people in the rest of the world. This parasitosis is still a neglected disease in which essential knowledge gaps persist regarding its pathogenesis, optimal treatment and prognostic factors. It is well known the relevance of the human microbiome and how significant changes in its composition can affect health. This is the consequence of the importance of microbial communities in immunological and biochemical functions. In this work, we have demonstrated significant changes in the microbiota of subjects with CD who exhibited consistent enrichment of *Parabacteroides spp* compared to healthy controls while for *Enterococcus hirae*, *Lactobacillus buchneri* and *Megamonas spp*, the effect was less clear once excluded the outliers values. On the other hand, sex, type of visceral involvement and previous treatment with benznidazole did not seem to have a role in gut microbiota structure. Given the current knowledge gaps in our understanding of CD pathogenesis, it will be essential to remain open-minded to other fields in biology.

## Introduction

*Trypanosoma cruzi* causes Chagas disease, a chronic infection in which the gut appears to be a reservoir from where this parasite triggers immune-mediated mechanisms which result in the damage of the enteric nervous systems and severe problems in intestinal motility [1–3]. Given the strong implications of the gut microbial communities in both the immune response against parasites and gut motility, the microbiota could explain the striking differences in clinical manifestations and treatment responses found in these patients.

This parasitosis is endemic in 21 countries in Latin America [4,5], where it causes 12,000 deaths per year. Estimations from 2010 show that nearly 6 million people are infected; most (62.4%) live in the Southern Cone, which has an at-risk population of 70.2 million people, and 38,593 new cases per year [6]. Due to globalisation and an increase in international migrations Chagas disease has also become a cause of concern in non-endemic countries, where up to 347,000[7] and 123,078[8] persons are infected in the United States and Europe, respectively. The most common route of transmission in endemic areas involves the contact with a *T. cruzi*-infected blood-sucking triatomine insect's urine or faeces through mucous membranes or non-intact skin [4,5,9]. Other modes of acquisition of the infection, which are also feasible in non-endemic areas, are transmission through blood and blood products or organ transplantation and congenital transmission [4,5]. The acute phase of *T. cruzi* infection if left untreated, progresses to a chronic phase in which 30–40% of infected individuals will develop visceral involvement, usually within 10–30 years. These chronic cases account for the vast majority of diagnoses outside endemic areas [10–12]. Around 15–45% of patients will develop cardiological manifestations, 10–21% will develop digestive involvement and 5–20% may have both [4,9,13–15]. Parasiticidal treatment of *T. cruzi* infection still relies on drugs licensed over 50

years ago: nifurtimox and benznidazole [16]. Their safety and efficacy profiles are far from ideal and are influenced principally by the infection's phase and the age of the patients. While the efficacy of benznidazole is high in the acute phase, congenital infections and reactivations in immunosuppressed individuals, cure rates drop significantly to 5–8% in the late chronic phase or in patients with cardiac involvement [16–18].

Digestive manifestations in Chagas disease seem to be related to the denervation of the enteric nervous system that may occur along the entire digestive tract, and causes severe alterations of the motility [1,19,20]. The exact mechanism of this denervation is still not entirely known, but immune mechanisms related to inflammation induced by the presence of the parasite may be involved [1]. It usually presents in the form of megavisceras: megacolon with or without oesophagopathy (70–87%), isolated oesophageal alterations (16–30%) and exceptionally biliary or small bowel dilatations [4,11,12,21], which can lead to severe constipation, intestinal obstruction, faecalomas, regurgitation, intestinal/esophageal perforation, or even esophageal carcinoma. However, important gaps remain related to the pathogenesis of chagasic gastrointestinal involvement, risk and prognostic factors, and response to parasiticidal treatment [22,23].

The microbiota can alter a parasite's colonization success, persistence, and virulence, shifting the parasitism-mutualism immune response from tolerogenic to inflammatory response [24]. The gut microbiota also plays a key role in gut motility [25]. For example, bacterial metabolites have shown to affect the excitability of the enteric and vagal afferent neurons that are damaged in Chagas disease [26]. In addition, mice with humanised microbiota from people with irritable bowel syndrome, developed alterations in intestinal transit and increased responses to pain [27].

Despite a strong rationale to expect an important role of the gut microbiota in Chagas disease establishment, clinical manifestations and treatment response, the impact of the microbiota on this disease remains barely studied, perhaps because it represents a neglected disease according to WHO [28]. Here, we aim to analyse the stool microbiota composition of patients with chronic Chagas disease with and without visceral involvement and compare it with non-infected controls. We also compared changes in the microbiota of patients treated vs not-treated with benznidazole.

## Materials and methods

### Ethics statement

The protocol was approved by the Ethics Committee of the Ramón y Cajal University Hospital (Ref. Acta 353; OCT/2018) and all participants signed an informed consent form.

### Study design, participants and data collection

We conducted a prospective cross-sectional study. Participants were recruited at the Tropical Medicine Referral Centres of the Ramón y Cajal University Hospital and La Paz University Hospital in Madrid, Spain, between February 2019 and October 2020. We included participants with chronic Chagas disease (diagnosed with two different serological tests), treated or not-treated with benznidazole, who were classified into three groups: patients with the indeterminate form (Indeterminate-CD), with only cardiac involvement (Cardiac-CD) and with only digestive involvement (Digestive-CD). We also included a control group of people from endemic areas with a negative screening for Chagas disease. We excluded patients with immunosuppression, <18 years of age and with significant heart or digestive comorbidities in addition to Chagas disease.

Participants attended a screening and a baseline visit where study procedures were performed. Stool were collected at the baseline visit (or within one week) and stored at -80˚C

until analysis. Study data were collected and managed using REDCap electronic data capture tools [29] hosted at Fundación SEIMC-GESIDA (https://fundacionseimcgesida.org/en).

## Food intake and dietary assessment

To determine food and beverage consumption, three 24-hour diet recalls were made over the course of a week on non-consecutive days one of which was a Sunday or holiday. The questionnaires were applied by trained personnel via telephone and using a three-step method, in which, initially, a quick list was made with only the foods or recipes indicated by the person interviewed. Subsequently, detailed questions were asked about all the foods that were part of the recipe or menu, including the type of food or beverage, quantity consumed and method of preparation; and finally, a review was made with the interviewee to clarify any ambiguities, ask if he/she took any medication or dietary supplement, and note the place where each meal was eaten, the time and the time spent on it. The recording format used was structured by meal (breakfast, mid-morning, lunch, snack, dinner, dinner or snack and other meals) [30,31]. Recognized measures and recipe ingredients were used to estimate portion sizes. The interviewer placed special emphasis on asking about foods consumed between meals or other frequently forgotten foods, such as bread, sugar, butter/margarine, sauces, etc.

All dietary information was processed with the DIAL software version 3.0.0.5 (Alce Ingeniería, Madrid, Spain) [32], which uses data from the Spanish Food Composition Tables [33]. The observed energy intake, the caloric profile of the macronutrients in the participants' diets, as well as the intake of vitamins and minerals were obtained through this program. The healthy dietary index [34] was also calculated, considering the specific dietary guidelines for the Spanish population [35,36].

## Sample collection and DNA extraction, library preparation, sequencing and bioinformatics analysis

**Samples collection and DNA extraction.** Fecal samples were stored in Omnigene Gut kits (DNA Genotek), which contain a stabilizer solution that better preserves (relative to RNA-later and Tris-EDTA) the composition of fecal microbial community structure DNA for microbiome analysis [37]. Fecal samples were aliquoted and cryopreserved at -80°C until use. Fecal DNA extraction were performed using QIAamp PowerFecal Pro DNA Kit (QIAGEN, Hilden, Germany)

## Library preparation and sequencing

The quality of input DNA was controlled with Nanodrop 2000 (Thermo Fisher Scientific, Waltham, MA) and concentration measured using Qubit 2.0 (Invitrogen by LifeTechnologies, Carlsbad, CA). Libraries for Whole Genome Sequence (WGS) were prepared following the protocol of Illumina DNA Prep, (M) Tagmentation kit and Nextera XT Index Kit v2 Set A (Illumina, San Diego, CA). Final fragment length distributions were determined using Tape Station 4150 (Agilent Technologies, Santa Clara, CA). The sequencing was performed using the kit NextSeq High Output (2x150 cycles) with NextSeq 500 sequencer (Illumina, San Diego, CA) at the Translational Genomics Unit. Hospital Universitario Ramón y Cajal, IRYCIS. Madrid, Spain.

**Preprocessing and quality control.** All the sequences used in this analysis passed quality control, where the length and quality of the reads were filtered using the *trimmomatic* v0.33 (Paired End method, minimum length of 100, average quality of 30) [38].

**Whole genome sequencing analysis.** WGS data was analyzed using the taxonomic sequence classifier Kraken (v2.0.7-beta, paired-end option) [39,40], which examines the k-mers within a query sequence and uses the information within those k-mers to query a

database. That database maps k-mers to the lowest common ancestor of all genomes known to contain a given k-mer. Taxonomic information on the WGS sequences was obtained using maxi-kraken database available in (https://lomanlab.github.io/mockcommunity/mc_databases.html) web. Abundance estimation was performed using Bracken software [41].

**Biodiversity and clustering.** Samples were rarified with the minimum sample classified reads in order to normalized the data among the samples. Alpha diversity metrics (Shannon and Simpson) were computed using the R package *vegan* [42] and compared using the *ggstatplot* R package [43]. Beta diversity was assessed using bray-curtis distances (R package *vegan*, function *vegdist*). Beta diversity distance was represented using UMAP algorithm (uwot R package). We applied the partial least squares discriminant analysis (sPLS-DA) using the *mixomics* R package [44] to further evaluate the differences in microbiota composition according to the Chagas disease status, a statistical method specially designed to handle high-dimensional, sparse data. We used cross-validation (*mixomics* package, function *perf)* to compute the evaluation criteria and fit an optimized sPLA-DA model restricted to the first 3 principal components and including 20, 6, and 20 features in components 1, 2 and 3, respectively. Association among beta diversity and variables were tested using PERMANOVA test (R package vegan, function adonis2). Biodiversity metrics were estimated considering all the taxonomic ranks.

**Differential abundance analysis.** Differential abundance tests were performed using DESeq2 package [45]. Input data was rarefied as previously described in order to reduce the false positive ratio. Significant level was stablished as <0.001 adjusted p-value. Volcano plots was performed using ggplot2. TreeMaps was performed by in house script. The heattree function is part of an R package under development. The code is accessible in the following github repository (https://github.com/irycisBioinfo/megalodon). A heat tree combines elements of a dendrogram and heatmaps to represent hierarchical clustering results. In the heat tree, each leaf node represents an individual observation, while internal nodes represent clusters formed by grouping similar observations. The colour of each leaf node indicates its value or membership strength. Examining the heat tree's branches and leaf nodes permits identifying patterns or relationships within the data.

## Data availability

The sequence data associated with this study have been deposited at EBI/ENA under accession number that will be provided upon manuscript publication.

## Statistical methods

We report qualitative variables as frequency distribution and quantitative variables as medians with their interquartile ranges. We performed comparisons between groups using the $\chi^2$ test for categorical variables. Since the distribution of all the assessed variables departed from normality after Shapiro Wilk tests, we used the Wilcoxon rank-sum test or the median test for the between-group comparisons of continuous variables.

# Results

## General characteristics of the study population

We included 60 patients with Chagas disease (29 I-CD, 16 C-CD and 15 D-CD) and 20 non-infected controls who did not differ significative from infected participants (Table 1).

Most of them were women of Bolivian origin in their forties with primary or secondary education levels. At baseline, all the patients had two positive serological results against *T. cruzi* and 30% also had a positive PCR for *T. cruzi*. No participant reported excessive alcohol consumption, and only one was a smoker (Indeterminate-CD). When we compared patients

Table 1. Baseline characteristics of study population.

| | Non-infected controls (n = 20) | Indeterminate CD (n = 29) | Cardiac CD (n = 16) | Digestive CD (n = 15) | Total (n = 80) | p value |
|---|---|---|---|---|---|---|
| Median age | 41 (27–49) | 47 (38–54) | 54 (44–58) | 49 (42–60) | 47 (39–54) | 0.18 |
| Sex (women) | 13 (65) | 22 (75.9) | 11 (68.7) | 15 (100) | 61 (76.2) | 0.09 |
| Country of origin | | | | | | 0.17 |
| Bolivia | 17 (85) | 27 (93.1) | 14 (87.5) | 14 (93.3) | 72 (90) | |
| Paraguay | 0 (0) | 2 (6.9) | 0 (0) | 1 (6.7) | 3 (3.7) | |
| Argentina | 2 (10) | 0 (0) | 0 (0) | 0 (0) | 2 (2.5) | |
| Peru | 1 (5) | 0 (0) | 0 (0) | 0 (0) | 1 (1.2) | |
| Brazil | 0 (0) | 0 (0) | 1 (6.2) | 0 (0) | 1 (1.2) | |
| Honduras | 0 (0) | 0 (0) | 1 (6.2) | 0 (0) | 1 (1.2) | |
| Education level | | | | | | 0.36 |
| No studies | 1 (5.0) | 4 (13.8) | 0 (0) | 0 (0) | 5 (6.2) | |
| Primary | 6 (30) | 11 (37.9) | 3 (18.7) | 8 (53.3) | 28 (35) | |
| Secondary | 11 (55) | 12 (41.4) | 11 (68.7) | 6 (40) | 40 (50) | |
| University/professional | 2 (10) | 2 (6.9) | 2 (12.5) | 1 (6.7) | 7 (8.7) | |
| Positive PCR for *T. cruzi**[*] | NA | 7 (28) | 2 (18.2) | 6 (42.9) | 15 (30) | 0.39 |

Data are numbers (%) or medians (interquartile range). [*]*Trypanosoma cruzi* PCR result was determined at the baseline visit or within the previous month for 50 patients out of 60 with Chagas disease.

CD: Chagas disease; NA: not applicable.

who received (n = 29) vs those who did not receive benznidazole treatment (n = 31), there were no significant differences across baseline characteristics (S1 Table). Six participants had severe and limiting visceral Chagas disease: three had cardiomyopathy (two dilated and one with an apical aneurysm), two had megacolon and one had achalasia. The remaining participants (n = 25) with determined Chagas disease had varying degrees of visceral involvement.

## Dietary quality assessment

Dietary data were obtained from 55 participants. After processing the dietary information, we detected no significant differences in dietary habits between patients with Chagas disease and controls in terms of both food groups and dietary components overall except for the consumption of sauces (Table 2) and the contribution to Dietary Reference Intake of vitamin K, which were higher in patients with Chagas disease (Table 3). When we compared the controls with the different groups of Chagas disease visceral involvement, we also found no significant differences between food consumption and daily intake of macronutrients and micronutrients in general, except in the energy provided by PUFAs and omega-6 fatty acids which were lower for those with digestive involvement and in the contribution to Dietary Reference Intake of vitamin K which was higher also in that group (S2 Table).

## Description of the diversity in gut bacterial communities

Alpha diversity measures the richness and evenness of bacterial taxa within a community. We found that bacterial richness was slightly higher in subjects with indeterminate Chagas disease, although the differences did not reach statistical significance (Fig 1).

Similarly, no clear differences were found at the family level. The most dominant family was the Ruminococcaceae, followed by Lachnospiraceae, Prevotellaceae and Bacteroidaceae (**Fig 2**).

**Table 2. Dietary data for participants with Chagas disease versus controls.**

| | Total (n = 55) | Chagas (n = 43) | Control (n = 12) | p value |
|---|---|---|---|---|
| Cereals (g/d) [t] | 147.9 ± 51.2 | 146.3 ± 51.2 | 153.8 ± 53.0 | 0.660 |
| Pulses (g/d) | 21.6 ± 43.8 | 23.6 ± 45.5 | 14.3 ± 38.1 | 0.309 |
| Greens and vegetables (g/d) [t] | 223.7 ± 118.4 | 225.7 ± 114.2 | 216.3 ± 137.4 | 0.811 |
| Fruits (g/d) | 200.6 ± 151.7 | 221.3 ± 159.1 | 126.7 ± 93.7 | 0.095 |
| Milk products (g/d) [t] | 272.1 ± 145.8 | 265.1 ± 141.9 | 297.1 ± 163.1 | 0.507 |
| Meat and meat products (g/d) [t] | 164.1 ± 72.1 | 157.6 ± 76.5 | 187.2 ± 49.5 | 0.212 |
| Fish and fish products (g/d) | 14.8 ± 19.6 | 14.9 ±19.5 | 14.3 ± 20.9 | 0.974 |
| Eggs (g/d) | 20.8 ± 23.6 | 22.7 ± 24.6 | 14.1 ± 18.8 | 0.213 |
| Sugars, sweets, pastries (g/d) | 34.2 ± 28.9 | 36.1 ± 31.2 | 27.4 ± 17.6 | 0.527 |
| Fat and oils (g/d) [t] | 33.8 ± 12.6 | 34.5 ± 13.6 | 31.3 ± 8.4 | 0.444 |
| Drinks (g/d) | 1249 ± 548 | 1263 ± 534 | 1202 ± 619 | 0.488 |
| Prepared and precooked foods (g/d) | 11.9 ± 22.2 | 10.1 ± 19.0 | 18.1 ± 31.2 | 0.569 |
| Aperitives (g/d) | 4.1 ± 11.2 | 2.4 ± 5.4 | 10.3 ± 21.3 | 0.835 |
| Sauces (g/d) | 15.9 ± 25.9 | 18.8 ± 28.0 | 5.6 ± 12.0 | 0.014 |
| Energy (kcal/d) | 1959 ± 358 | 1935 ± 365 | 2044 ± 332 | 0.299 |
| Energy (%EE) [t] | 94.1 ± 8.9 | 93.8 ± 9.7 | 95.1 ± 5.8 | 0.670 |
| Proteins (g/d) | 75.4 ± 19.1 | 75.2 ± 20.1 | 76.7 ± 15.8 | 0.639 |
| Proteins (%TEI) [t] | 15.5 ± 3.2 | 15.6 ± 3.3 | 15.1 ± 2.6 | 0.626 |
| Carbohydrates (g/d) [t] | 206.4 ± 47.8 | 201.1 ± 46.3 | 225.6 ± 50.1 | 0.117 |
| Carbohydrates (%TEI) [t] | 42.2 ± 6.7 | 41.7 ± 7.0 | 44.0 ± 5.8 | 0.304 |
| Fiber (g/d) | 18.1 ± 5.9 | 18.9 ± 5.9 | 15.0 ± 5.1 | 0.087 |
| Lipids (g/d) [t] | 87.2 ± 21.7 | 86.9 ± 23.3 | 88.3 ± 15.5 | 0.844 |
| Lipids (%TEI) [t] | 40.0 ± 5.6 | 40.2 ± 5.9 | 39.0 ± 4.4 | 0.501 |
| SFA (%TEI) [t] | 11.9 ± 2.9 | 11.8 ± 3.0 | 12.5 ± 2.6 | 0.419 |
| PUFA (%TEI) | 6.8 ± 2.6 | 6.7 ± 2.6 | 7.1 ± 2.4 | 0.392 |
| MUFA (%TEI) [t] | 17.6 ± 4.1 | 18.1 ± 3.9 | 15.8 ± 4.3 | 0.077 |
| Omega 3 fatty acids (%TEI) | 0.67 ± 0.36 | 0.70 ± 0.38 | 0.56 ± 0.24 | 0.221 |
| Alpha linolenic acid (%TEI) | 0.53 ± 0.32 | 0.56 ± 0.35 | 0.42 ± 0.11 | 0.166 |
| EPA + DHA (mg/d) | 222.6 ± 299.3 | 244.6 ± 331.1 | 143.9 ± 111.9 | 0.476 |
| n 6 fatty acids (%TEI) | 6.0 ± 2.5 | 5.8 ± 2.5 | 6.5 ± 2.4 | 0.254 |
| Omega 6 (g/d)/Omega 3 (g/d) ratio | 11.0±7.8 | 10.4±8.1 | 13.0±6.5 | 0.083 |
| Trans fatty acids (%TEI) [t] | 0.41 ± 0.21 | 0.39 ±0.21 | 0.47 ± 0.16 | 0.274 |
| Cholesterol (mg/d) [t] | 283.4 ± 96.2 | 283.0 ± 103.4 | 284.7 ± 67.9 | 0.959 |
| Alcohol (g/d) | 1.5 ± 4.4 | 1.5 ± 4.5 | 1.5 ± 3.9 | 0.459 |
| Healthy Eating Index [t] | 63.5 ± 11.8 | 64.1 ± 12.7 | 61.1 ± 7.6 | 0.435 |

Results are presented as mean ± SD. Significant differences between dietary patterns. (t): p-value calculated with t-test for independent samples. Other variables: p-value calculated with Mann–Whitney U test.

food = g edible per day; EE: Energy expenditure; TEI: Total energy intake; SFA: Saturated fatty acids; PUFA: Polyunsaturated fatty acids; MUFA: Monounsaturated fatty acids; EPA: eicosapentanoic acid; DHA: docosahexaenoic acid.

Next, we assessed differences in overall microbiota structure by analyzing beta-diversity to detect sample clustering. Using unsupervised UMAP analysis, we found no differences according to Chagas disease status (Fig 3A). A PERMANOVA analysis to test differences in beta diversity between Chagas disease groups did not suggest a significant effect (R2 = 0.047, P = 0.153). However, sPLS-DA analysis—an approach that better deals with sparse data—, indicated that 5% of the observed variability of the microbiota was explained by the presence or absence of Chagas disease (Fig 3B and 3C), indicating that the disease condition exerts a relevant impact on microbiota composition. The sPLS-DA analysis used to quantify this 5% effect of Chagas' disease on microbiota composition showed a high discriminative performance (AUC 0.996, Fig 3C).

We further investigated which genera determined divergences of microbial communities across subjects with and without Chagas disease by identifying in Volcano plots the most differentially abundant in each group (Fig 4). *Lactobacillus buchneri* and *Enterococcus hirae* were

**Table 3. Contribution to the recommended daily intakes of vitamins and minerals for participants with Chagas disease versus controls.**

| %DRI | Total (n = 55) | Chagas (n = 43) | Control (n = 12) | p value |
|---|---|---|---|---|
| Thiamine | 103.5 ± 28.0 | 104.6 ± 27.6 | 99.4 ± 30.1 | 0.482 |
| Riboflavin | 116.2 ± 33.6 | 117.4 ± 34.0 | 111.7 ± 33.1 | 0.767 |
| Vitamin B6 [(t)] | 141.0 ± 40.8 | 137.0 ± 36.9 | 155.1 ± 52.0 | 0.176 |
| Vitamin B12 | 155.1 ± 67.3 | 157.4 ± 68.6 | 146.9 ± 64.6 | 0.501 |
| Niacin | 197.9 ± 43.1 | 194.3 ± 39.6 | 210.5 ± 53.9 | 0.338 |
| Folic acid [(t)] | 54.9 ± 18.2 | 57.3 ± 18.4 | 46.1 ± 15.0 | 0.058 |
| Vitamin C [(t)] | 185.9 ± 89.7 | 192.8 ± 92.4 | 159.9 ± 78.3 | 0.258 |
| Pantothenic acid [(t)] | 95.6 ± 22.3 | 94.0 ± 21.5 | 101.7 ± 25.1 | 0.294 |
| Biotin [(t)] | 84.1 ± 39.6 | 87.2 ± 42.1 | 72.9 ± 27.7 | 0.273 |
| Vitamin A | 88.1 ± 45.3 | 92.8 ± 45.9 | 71.2 ± 40.2 | 0.087 |
| Vitamin D | 10.3 ± 11.4 | 11.4 ± 12.3 | 6.5 ± 6.5 | 0.107 |
| Vitamin E | 124.9 ± 61.0 | 125.0 ± 64.4 | 124.3 ± 49.2 | 0.596 |
| Vitamina K | 120.4 ±77.5 | 131.5 ± 77.1 | 80.6 ± 67.5 | 0.014 |
| Calcium [(t)] | 56.1 ± 20.2 | 55.0 ± 19.4 | 59.9 ± 23.2 | 0.464 |
| Phosphorus [(t)] | 172.1 ± 42.9 | 171.0 ± 40.4 | 176.3 ± 50.3 | 0.705 |
| Magnesium [(t)] | 70.0 ± 18.8 | 71.6 ± 19.6 | 63.8 ± 15.1 | 0.206 |
| Iron | 99.8 ± 34.1 | 99.1 ± 30.4 | 102.4 ± 46.7 | 0.879 |
| Zinc [(t)] | 69.7 ± 15.5 | 71.0 ± 15.9 | 65.0 ± 13.3 | 0.240 |
| Iodine [(t)] | 49.1 ± 18.7 | 48.7 ± 17.8 | 50.4 ± 22.5 | 0.778 |
| Selenium [(t)] | 127.7 ± 35.7 | 124.1 ± 30.7 | 140.3 ± 49.4 | 0.167 |

Results are presented as mean ± SD. (t): p-value calculated with t-test for independent samples. Other variables: p-value calculated with Mann–Whitney U test.

DRI: Dietary Reference Intakes.

the most enriched species in subjects with Chagas disease, who also exhibited several species belonging to the *Megamonas* genus. In contrast, the most depleted family in patients with Chagas disease was *Spirochaetaceae* and especially the genus *Treponema* with several species (*T. succinifaciens, T. porcinum and T. briantii*) was significantly more abundant in control subjects. We also found *Salmonella enterica* was relatively more abundant in control participants compared to patients with Chagas disease.

Then, to avoid inferring as significant the differences driven by extreme values in single individuals, we inspected the relative abundance of each of the taxa revealed as potentially relevant in the previous step (Fig 5). We found that subjects with Chagas disease exhibited enrichment of *Enterococcus hirae, Lactobacillus buchneri, Megamonas spp.* and *Parabacteroides spp.*, while differences in *Treponema spp.* appeared less clear. The patients most enriched in *E. hirae, Lactobacillus spp* and *Megamonas spp* were those with Chagas disease (as opposed to controls), regardless of the form of the disease (indeterminate, cardiac or digestive) and its severity. While those outliers were present in any form of the disease for *E. hirae*, they were more common in the digestive form for *Megamonas spp* and were not present in the digestive form for *Lactobacillus spp*. However, in consecutive Mann Whitney's U tests to minimize the impact of outliers, only *Parabacteroides spp.* retained the statistical significance (p = 0.038).

Finally, to assess the potential confounding effects of sex and previous Chagas treatment on gut microbiota structure, we represented in a heatmap the relative abundance of each of the bacteria identified as a biomarker of Chagas disease in relation to these potential confounders, which did not appear to form clusters (Fig 6). In addition, none of these variables was significant in a PERMANOVA analysis including the study group, Chagas treatment and sex as covariates.

Lastly, because of the potential implications of the microbiota on gut motility, in an exploratory subanalysis we investigated the bacterial biomarkers of GI Chagas disease (Fig 7). We

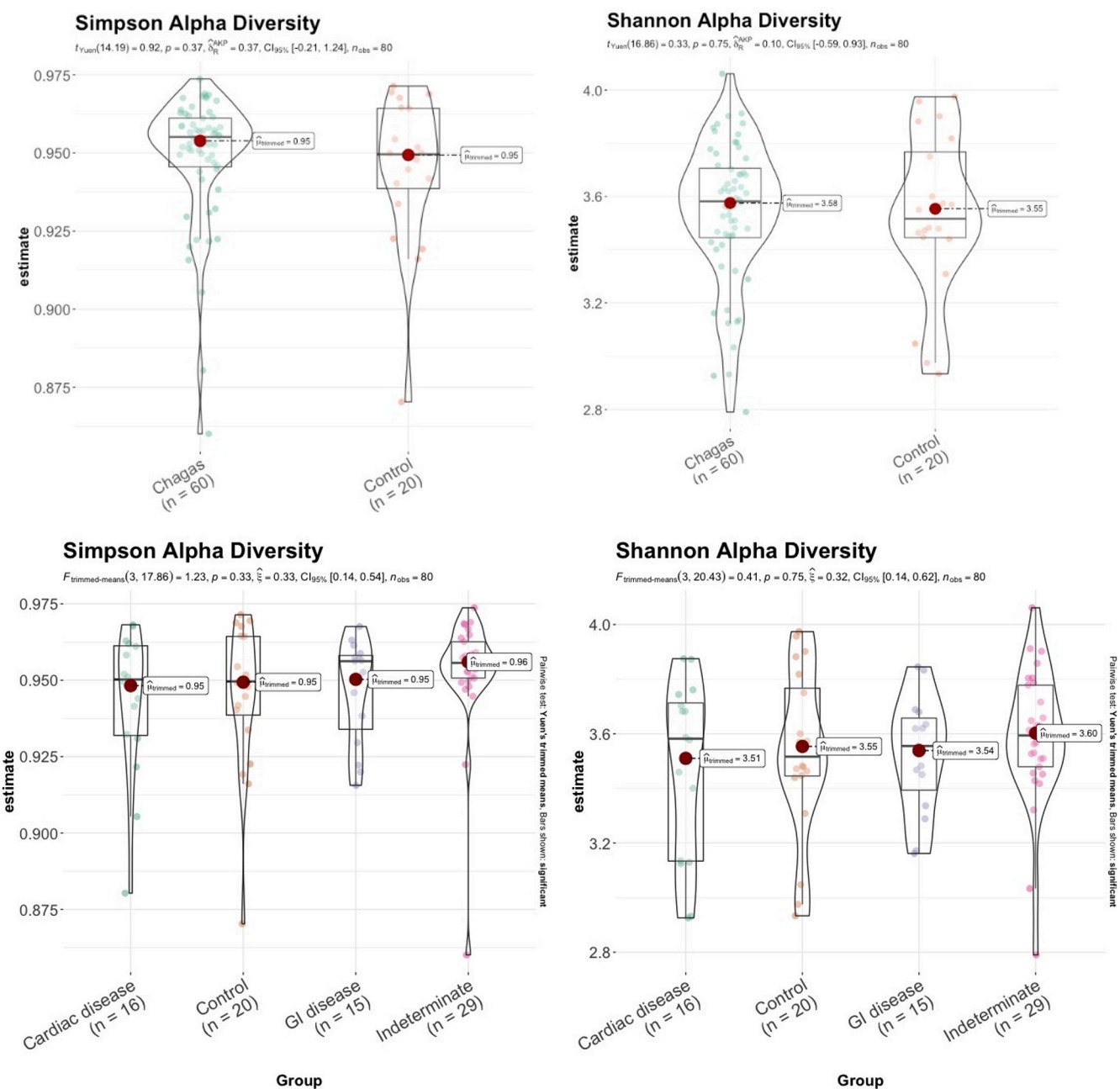

**Fig 1. Bacterial richness (Shannon and Simpson indices) according to the presence of Chagas disease and the presence of visceral involvement.**

found that *Enterococcus hirae* was depleted in subjects with GI Chagas disease, while *Prevotella sp*. were the most consistently enriched.

## Discussion

In this characterization of the microbiota in patients with Chagas disease, we found no impact of this parasitosis on bacterial richness, but a detectable effect on overall microbiota structure (5% of the microbiota variability explained by Chagas condition) with several biomarkers. In our study some taxa showed predominance in the gut microbiome of the entire studied

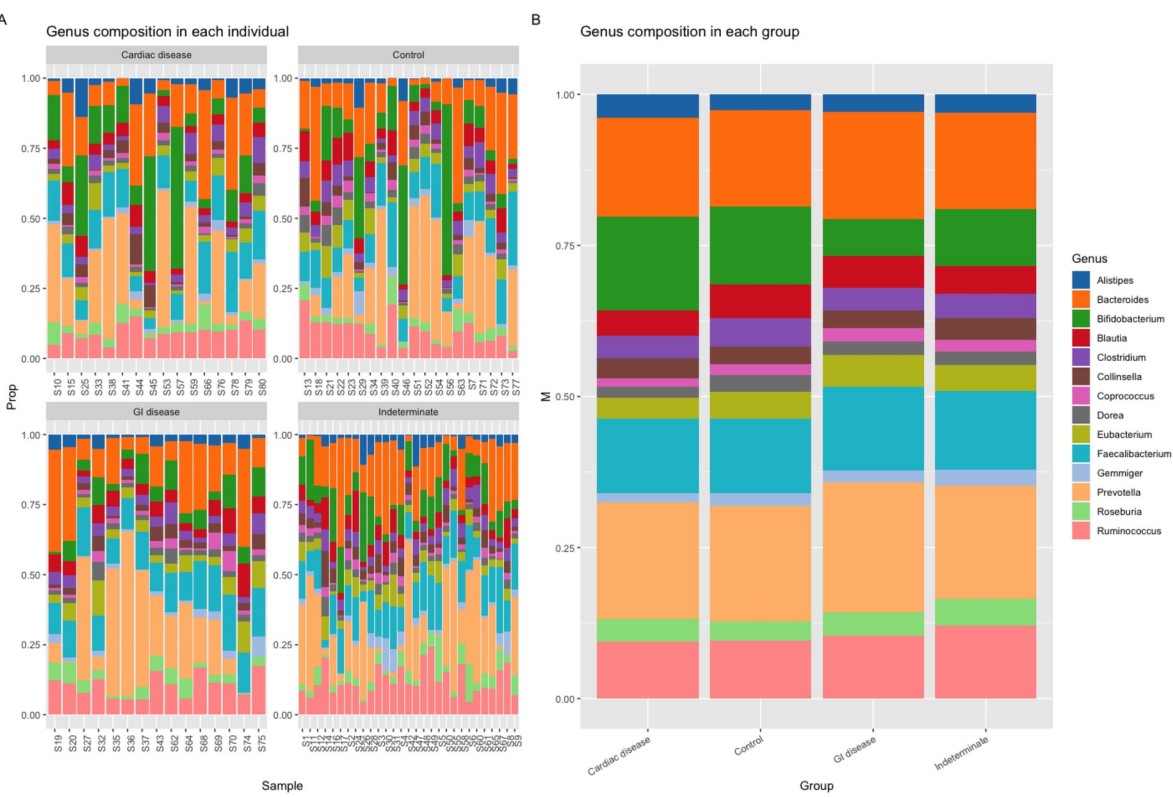

**Fig 2. Barplots representing the top 14 most abundant bacteria at the family level in each individual (left panel) and in each group (right panel).**

population such as *Ruminococcaceae*, *Prevotella* and *Bacteroides*. This characteristic composition has been described in other studies in migrants of Latin American origin in which *Ruminococcoccaceae* was more abundant and the gut microbiome was characterised by a relatively high proportion of *Prevotella* to *Bacteroides* [46]. Part of the relative diversity detected was explained by more favourable dietary habits, with higher consumption of fiber, a lower intake of red meat, and lower trans fats comsumption. Although in our study the mean fibre intake was lower than the Adequate Intake established by the EFSA of 25 grams per day [47] or that recommended by the Institute of Medicine [48], it was still higher than that observed in a sample of 1655 Spanish adults aged 18–64 years in the ANIBES study (12.5 g/d) [49]. A diet rich in fibre in our study participants would justify a high proportion of *Ruminococcaceae*, since this family is highly specialized in the degradation of complex plant material to be converted into short chain fatty acids (mainly acetate, butyrate, and propionate) that can be absorbed and used by the host [50].

In our study, Chagas disease explained 5% of the variability of the microbiota. This effect size is consistent with many biologically-meaningful effects on the gut microbiota identified in large studies powered to find small differences [51,52]. Mechanistically, multiple factors can explain reciprocal interactions between the microbiota and Chagas disease. The enteric nervous system controls intestinal motility and secretion quite autonomously from the central nervous system, sympathetic and parasympathetic systems. Digestive manifestations in Chagas disease seem to be related to the denervation of the enteric nervous system, that may occur along the entire digestive tract, and causes severe alterations of the motility [1,19,20], potentially impacting the microbiota. The exact mechanism of this denervation is still not entirely

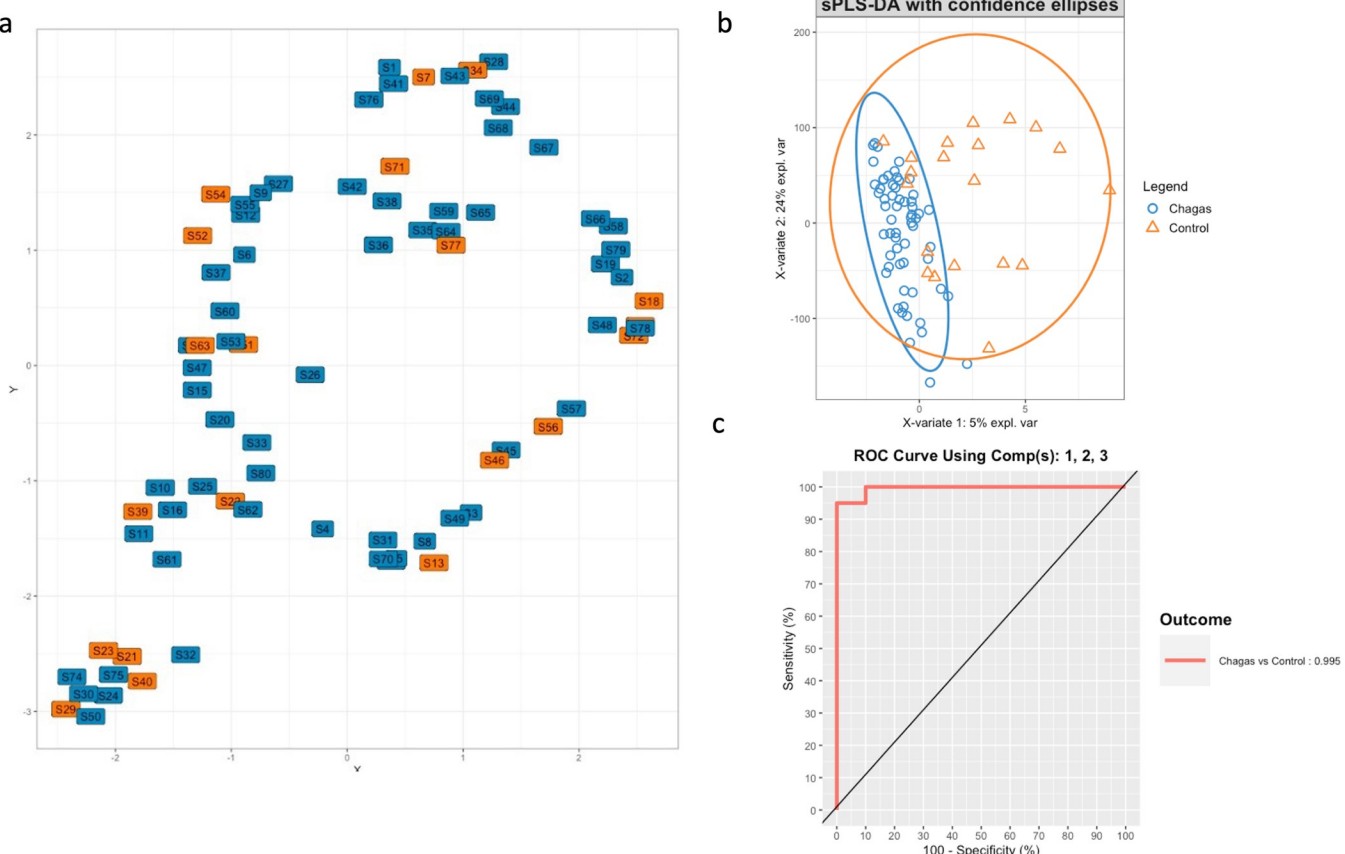

**Fig 3.** (**a**) Unsupervised clustering of beta diversity by UMAP analysis. (**b**) Explained variance of microbiota composition in patients with Chagas diseases vs. controls using sPLS-DA modelization. The plot use the first two components as axes and shows that the variance explained by the disease group occurs in the X axis and corresponds to a 5%. The graph depicts the samples with the confidence ellipses of different class labels, (**c**) depicts the area under the Receiver Operating Characteristic Curve of the optimized sPLS-DA model for the effect of Chagas disease on the microbiota composition.

known, but immune mechanisms related to inflammation induced by the presence of the parasite may be involved [1]. In addition, the gut may act as the primary reservoir for *T. cruzi* in the chronic phase, suggesting that local infection could potentially influence the development of digestive disease and could also serve as a reservoir for parasites involved in Chagas heart disease pathogenesis [2,3].

In addition, because of the central role of the gut bacterial communities in shaping the immune responses [53], the microbiota could explain the differing clinical consequences of Chagas disease between individuals. For example, the gut microbiota can directly stimulate enteric neurons and immune cells and affect intraluminal metabolism [54], explaining why it is now considered a key player in gut motility [25]. In animal models, the microbiota metabolic derivatives have been shown to affect the excitability of enteric and vagal afferent neurons [26]. In rats free of micro-organisms, profound alterations of intestinal motility occur, which can be modified by colonisation with bacteria such as *Lactobacillus acidophilus*, *Bifidobacterium bifidum* or *Micrococcus luteus* [55]. It has also been shown that mice with humanised microbiota from people with irritable bowel syndrome, developed alterations in intestinal transit and increased responses to pain [27].

The potential impact of Chagas disease on the gut microbiome has been analyzed previously in a sample of the Brazilian adult population [56]. The stool microbiome of 104

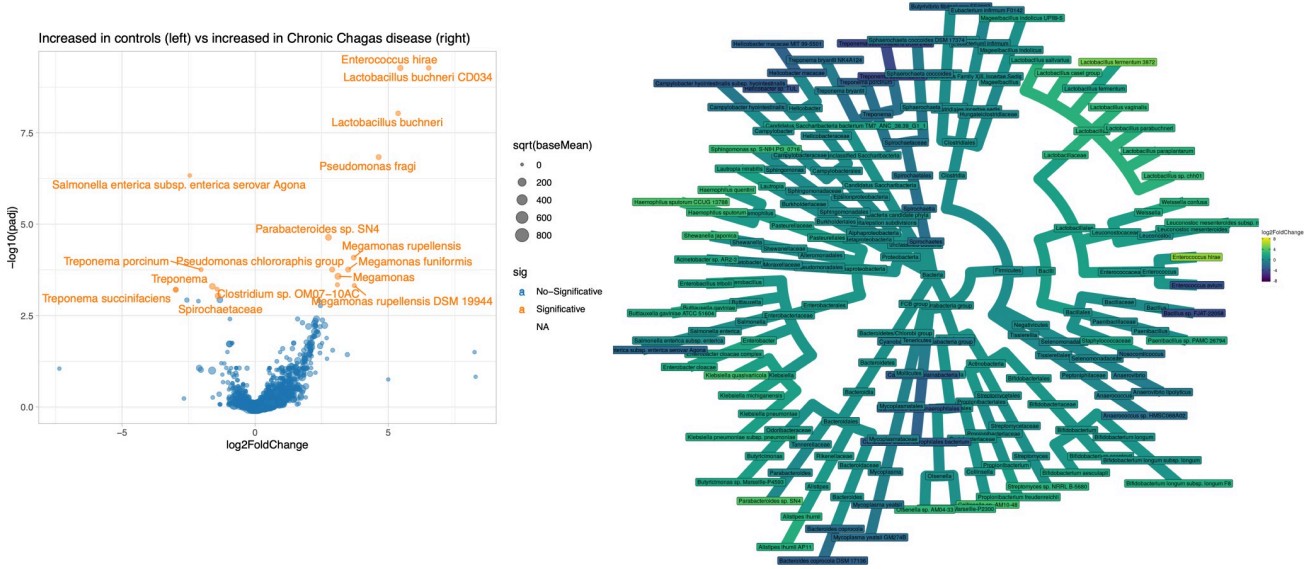

**Fig 4. Volcano plot and Heat tree showing the differential abundance of species between controls and patients with Chagas disease.**

individuals, 73 with Chagas disease (with the cardiac, digestive and indeterminate form) and 31 healthy controls, was characterized using 16S rRNA amplification and sequencing. The authors found that the genus *Akkermansia* was significantly lower in patients with Chagas disease, especially the cardiac group, compared to the controls. *Akkermansia* is a butyrate-producing bacteria associated with a healthy gut and has been related to decreased inflammation in animal studies [57]. In addition, differences in the relative abundances of *Alistipes*, *Bilophila*, and *Dialister* were observed between the groups, being more common in patients with cardiac Chagas disease. Those genera have been related to bowel inflammation, animal-based diets and diabetes [58–60]. In this study, *T. cruzi* infection was associated with changes in the gut microbiome that may play a role in the myocardial and intestinal inflammation seen in Chagas disease. The differences detected in the microbiota characterization between the study of de Souza-Basqueira and ours may be explained by some substantial differences between the two studies, such as the population included (in our case, mainly Bolivians), the endemicity of *T. cruzi* in the participants' region of residence or the type of diet.

In addition, there is some evidence that the alterations in the microbiome could be restored by treatment with benznidazole. In a study of children with and without chronic Chagas disease, the infected children presented higher fecal *Firmicutes* (*Streptococcus*, *Roseburia*, *Butyrivibrio*, and *Blautia*) and lower *Bacteroides* concentrations. Also, they showed some skin (but not oral) microbiota differences. Treatment with benznidazole eliminated the fecal microbiota differences but not the skin and oral ones [61]. In our study, we did not find a clear impact of Chagas treatment on gut microbiota composition, although we could not assess the impact within individuals due to the cross-sectional design. Furthermore, in a murine model, the infection by *T. cruzi* caused joint microbial and chemical perturbations, including alterations in conjugated linoleic acid derivatives and in specific members of families *Ruminococcaceae* and *Lachnospiraceae*, as well as alterations in secondary bile acids and members of order *Clostridiales* [62].

We found a significant enrichment of *Parabacteroides spp* among patients with Chagas disease. This genus has been associated with various diseases and conditions, including metabolic disorders, autoimmune diseases, and gastrointestinal disorders [63,64]. It is possible that *Parabacteroides spp* plays a role in modulating the immune response or altering the gut barrier

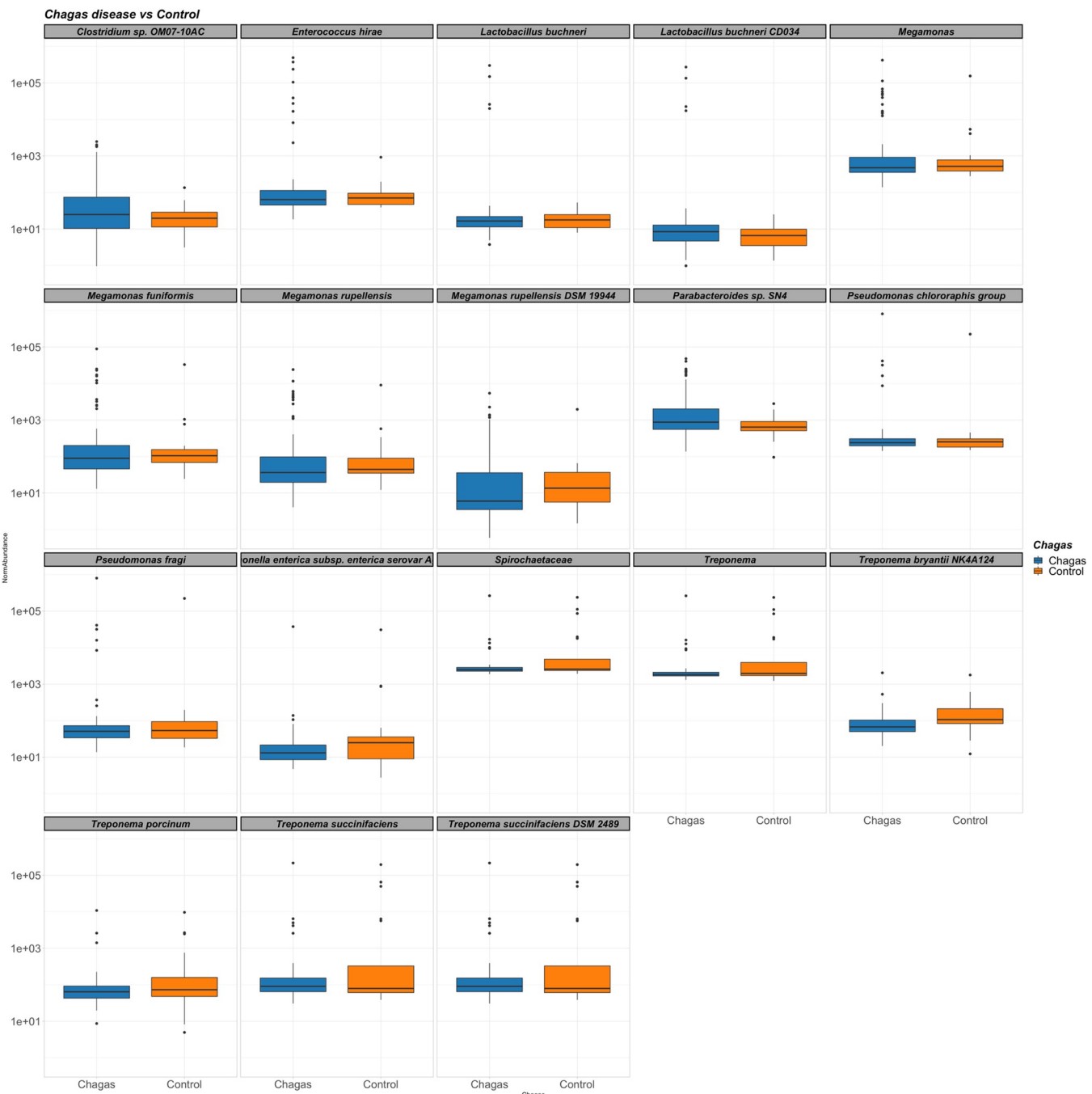

**Fig 5. Boxplots depicting the relative abundance of the bacterial biomarkers of Chagas disease identified by Deseq2 analysis.** When computing the statistical significance using Mann Whitney's U tests, only the differences Parabacteroides spp. remained statistically significant (p = 0.038).

function in Chagas disease patients. In addition, *Parabacteroides spp* have been associated with the production of short-chain fatty acids (SCFAs) such as propionate and acetate [64]. These SCFAs can influence host metabolism and immune responses. It would be interesting to investigate whether the increased abundance of *Parabacteroides spp* in Chagas disease patients is associated with altered SCFA production and if this contributes to the pathogenesis of the disease.

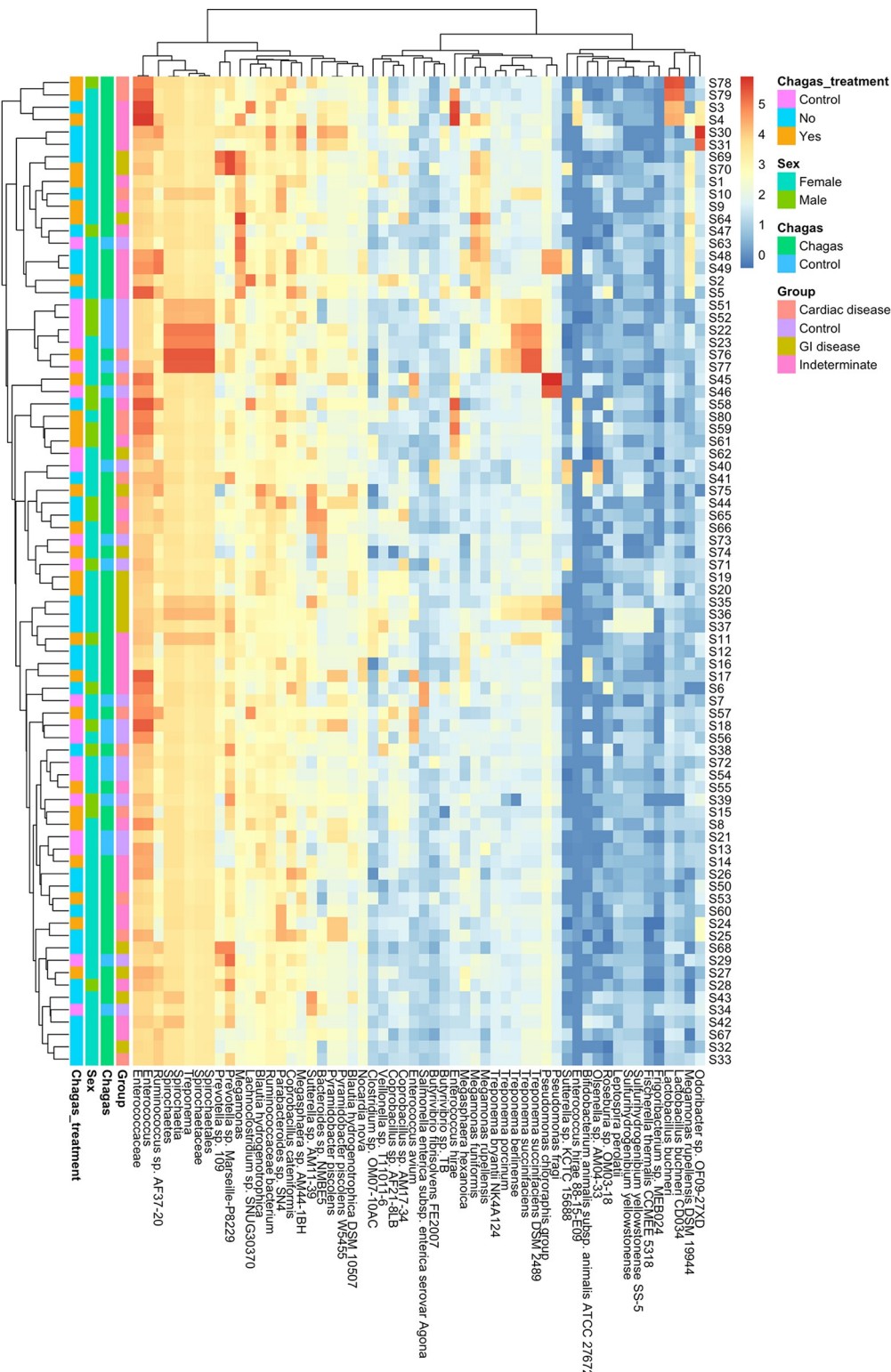

**Fig 6. Heatmap of the relative abundance of species differentially abundant in subjects with Chagas disease and hierarchical clustering.** Sex, previous Chagas treatment and study groups are annotated in columns.

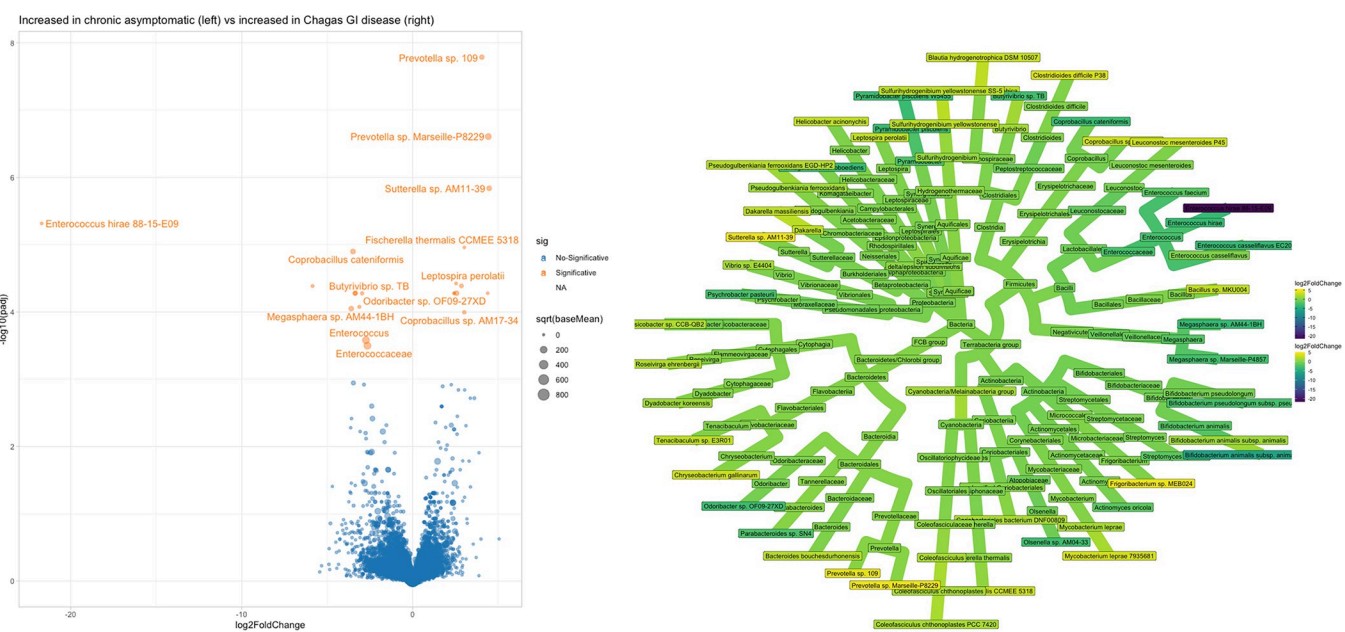

**Fig 7. Volcano plot and Heat tree showing the differential abundance of species between patients with chronic indeterminate disease and those with GI.**

We also detected that subjects with Chagas disease exhibited enrichment of *Enterococcus hirae*. This family is constituted by commensal bacteria with a well-adapted mechanism to survive in the gastrointestinal tract of humans, animals and insects where they contribute to digestion and gut metabolic pathways [65]. Although foodborne *Enterococcus spp.* are rarely implicated as pathogens, consumption of these bacteria can lead to their establishment in the gastrointestinal tract [66]. While *E. faecium* and *E. faecalis* are more prevalent in human-associated environments, *E. hirae* is a common coloniser of animal species, especially cattle, and can be easily isolated from cattle manure and water samples from feeding basins [67,68]. *E. hirae* was the predominant species recovered from cattle production systems including both bovine feces (92%), and feedlot catch basins (60%), it was not isolated in any of the 1849 human clinical samples it was sampled [68]. *E. hirae* has also been described in psittacine birds and chickens [69] and in rats and cats [70]. Human *E. hirae* infections are very rare and opportunistic in nature. Reported cases include urinary tract infections, biliary tract infections, infective endocarditis and catheter-related bloodstream infections [71]. In pregnant women with bacterial vaginosis, *E. hirae* has been involved as a marker of the disbalance of the vaginal ecosystem, being negatively correlated with a normal Nugent rating [72].

Such niche-specificity for *E. hirae* and its rare presence in the commensal flora of human beings makes this bacterium an attractive indicator for further investigation. Given the lack of differences in the geographic origin and the lack of substantial differences in the dietary patterns between our study groups, we do not think that the abundance of *E. hirae* could have been confounded by these factors. However, we recognise that part of the effect may be due to enrichment outliers observed with this species. Therefore, in patients with Chagas disease, it may be a marker of altered bacterial homeostasis when Chagas disease has not yet caused severe impairment of intestinal structure and motility, which is when the presence of *Prevotella spp or Parabacteroides spp*. could act as an alert.

*Prevotella spp* was another relevant genus since it was enriched in subjects with gastrointestinal Chagas disease. *Prevotella spp*, a dominant genus in the Bacteroidetes class, includes more

than 50 species, mostly found in humans, and are considered key players in the balance between health and disease [73]. *Prevotella spp* is considered a pro-inflammatory bacteria, following studies showing detrimental effects in rheumatoid arthritis [74], ankylosing spondylitis [75], or in people with HIV where it correlates with mucosal and systemic immune activation [76].

As for patients with digestive involvement, their intake of PUFA and omega-6 fatty acids was lower compared to the group with indeterminate Chagas disease, while the omega-6/omega-3 ratio was similar across all groups (S2 Table). It has been pointed out that due to the anti-inflammatory effect of omega-3 and the pro-inflammatory effect of omega-6, it is more precise to analyse the ratio of both fatty acids in the diet rather than their intake separately [77].

The main strength of our study is novelty, given the scarcity of studies on the characterization of the microbiota in subjects living in Chagas disease non-endemic areas, with different degrees of visceral involvement, compared to unaffected individuals. Another strength of our study is the careful dietary assay, which allowed us to address the potential role of diet as a confounding factor. Our study is also subject to several limitations. The main limitation relies on its cross-sectional design. This prevented us from assessing cause-effect relationships among microbiota composition and either presence or absence of Chagas disease or Chagas disease grade of visceral involvement. The small sample size prevented us from performing multiple subgroup analyses to reduce the risk of false discoveries. Therefore, the primary analysis focused on the two main groups of interest (chagasic versus non-chagasic) with some data on the group with potentially more altered microbiota (those with GI involvement). Lastly, we could not investigate the mechanisms by which the discriminative bacteria could influence Chagas disease. We are planning to perform additional techniques, including pathway analyses and metabolomics to provide further mechanistic insight.

Our findings encourage further research in this field. Future studies could focus on better understanding the cause-effect relationship between human susceptibility to *T. cruzi* infection, the progression of Chagas disease, and the response to parasiticidal treatments. Given the current knowledge gaps in our understanding *T. cruzi* pathogenesis, it will be important to remain open-minded to other fields in biology. The potential rewards are important: the microbiota could prove to be a viable target to improve the prognosis of this prevalent and neglected disease.

## Supporting information

**S1 Table. Baseline characteristics of study population according to treatment.**
(DOCX)

**S2 Table. Dietary data for participants with Chagas disease versus controls.**
(DOCX)

## Author Contributions

**Conceptualization:** José A. Pérez-Molina, Clara Crespillo-Andújar, Sergio Serrano-Villar.

**Data curation:** José A. Pérez-Molina, Leticia Olavarrieta, Beatriz Navia, Val F. Lanza, Sergio Serrano-Villar.

**Formal analysis:** José A. Pérez-Molina, Leticia Olavarrieta, Beatriz Navia, Val F. Lanza, Sergio Serrano-Villar.

**Funding acquisition:** José A. Pérez-Molina.

**Methodology:** José A. Pérez-Molina, Clara Crespillo-Andújar, Leticia Olavarrieta, Beatriz Navia, Val F. Lanza, Sergio Serrano-Villar.

**Project administration:** José A. Pérez-Molina, Clara Crespillo-Andújar, Elena Trigo, Sergio Serrano-Villar.

**Resources:** José A. Pérez-Molina, Sergio Serrano-Villar.

**Software:** José A. Pérez-Molina, Val F. Lanza, Sergio Serrano-Villar.

**Visualization:** José A. Pérez-Molina, Val F. Lanza, Sergio Serrano-Villar.

**Writing – original draft:** José A. Pérez-Molina, Sergio Serrano-Villar.

**Writing – review & editing:** José A. Pérez-Molina, Clara Crespillo-Andújar, Elena Trigo, Sandra Chamorro, Marta Arsuaga, Leticia Olavarrieta, Beatriz Navia, Oihane Martín, Begoña Monge-Maillo, Francesca F. Norman, Val F. Lanza, Sergio Serrano-Villar.

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
