## [Decision Letter · Decision Letter 0]

9 Apr 2023

Dear PhD Pérez-Molina,

Thank you very much for submitting your manuscript "Chagas disease is related to structural changes of the gut microbiota in adults with chronic infection (TRIPOBIOME Study)" for consideration at PLOS Neglected Tropical Diseases. As with all papers reviewed by the journal, your manuscript was reviewed by members of the editorial board and by several independent reviewers. In light of the reviews (below this email), we would like to invite the resubmission of a significantly-revised version that takes into account the reviewers' comments. 

I am really sorry for the delay. 

Unfortunately, your manuscript was not handled in an expedient manner. The Academic Editor did not reply to the staff e-mails, and I wasn't cc on this correspondence. 

In the mean time, I am sending you the comments we received from the lone reviewer. Upon receipt of your revised manuscript I will handle it personally, and send it out for review.

We cannot make any decision about publication until we have seen the revised manuscript and your response to the reviewers' comments. Your revised manuscript is also likely to be sent to reviewers for further evaluation.

Sincerely,

Charles L. Jaffe, Ph.D.

Section Editor

Charles Jaffe

Section Editor

I am really sorry for the delay. 

Unfortunately, your manuscript was not handled in an expedient manner. The Academic Editor did not reply to the staff e-mails, and I wasn't cc on this correspondence. 

In the mean time, I am sending you the comments we received from the lone reviewer. Upon receipt of your revised manuscript I will handle it personally, and send it out for review.

Reviewer's Responses to Questions

**Key Review Criteria Required for Acceptance?**

**Methods**

-Are the objectives of the study clearly articulated with a clear testable hypothesis stated?

-Is the study design appropriate to address the stated objectives?

-Is the population clearly described and appropriate for the hypothesis being tested?

-Is the sample size sufficient to ensure adequate power to address the hypothesis being tested?

-Were correct statistical analysis used to support conclusions?

-Are there concerns about ethical or regulatory requirements being met?

Reviewer #1: see below

**Results**

-Does the analysis presented match the analysis plan?

-Are the results clearly and completely presented?

-Are the figures (Tables, Images) of sufficient quality for clarity?

Reviewer #1: see below

**Conclusions**

-Are the conclusions supported by the data presented?

-Are the limitations of analysis clearly described?

-Do the authors discuss how these data can be helpful to advance our understanding of the topic under study?

-Is public health relevance addressed?

Reviewer #1: see below

**Editorial and Data Presentation Modifications?**

Reviewer #1: see below

**Summary and General Comments**

Reviewer #1: In this manuscript, Pérez-Molina et al performed fecal microbiome characterization of Chagas disease patients and controls. Overall, this is a well-designed study that uses appropriate methods and controls. A particular strength of this study is the dietary questionnaire, which enables the exclusion of dietary confounders. However, authors over-interpret minor differences, and make statements that are not supported by the data. Specifically, I have the following major concerns:

1. PLS-DA is prone to over-fitting. Authors should provide metrics to confirm that this is not the case here (explained variation (R2) and predictive ability (Q2)). 

2. Heat tree analyses are an atypical way to present the data. Authors should provide additional text on figure interpretations, and on the methods used to generate this figure.

3. Authors should also clarify the discrepancies between the differential bacteria displayed on the volcano plot and on the heat tree.

4. Lines 275-277: this statement is not supported by the data. Indeed, the differences in Enterococcus hirae, Lactobacillus buchneri, and Megamonas between CD and controls appear to be primarily outlier-derived, with group medians and quartile overlapping fully. The differences in Parabacteroides appear more robust.

5. Authors should test whether these outliers come from the six patients with more-severe disease symptoms

6. Authors use the heatmap in figure 6 to justify a lack of impact of dietary, treatment or sex-based confounders. However, given that the samples also do not cluster by Chagas status, the study may simply be under-powered to address confounders. 

7. Authors under-used their metagenomics data by merely performing species-level analyses. Important analyses should also investigate whether there are specific bacterial genes or pathways that differ between groups. 

8. Authors fail to cite recent work already published on this topic using 16S sequencing (de Souza-Basqueira et al, doi: 10.3389/fcimb.2020.00402). A comparison of findings from the current study vs this prior publication should be performed. This also leads to the incorrect statement of novelty by the authors at lines 387-389.

9. Lines 304-305: “we found a small impact of this parasitosis on bacterial richness”. Richness differences did not reach statistical significance and thus should not be stated as a finding.

10. Authors should specify whether beta-diversity differences were observed between the different CD forms

Minor concerns: 

1. Line 223-225: this statement is not supported by Table S2, since significant differences in PUFA and Omega 6 fatty acids were observed.

2. Line 223 (“data not shown”). Data should be provided in SI.

3. There are two color legends in Figure 4b. 

4. Line 113-118: the details on REDcap are unnecessary and irrelevant to the study’s findings

5. Given that E. hirae is rarely associated with humans, and yet is found at relatively high abundance in both CD and controls in this study, authors should verify their taxonomic assignments, before making this large interpretation.

6. Lines 380-385: italicize Prevotella.

PLOS authors have the option to publish the peer review history of their article (what does this mean?). If published, this will include your full peer review and any attached files.

Reviewer #1: No
---

## [Decision Letter · Decision Letter 1]

30 Jun 2023

Dear PhD Pérez-Molina,

We are pleased to inform you that your manuscript 'Chagas disease is related to structural changes of the gut microbiota in adults with chronic infection (TRIPOBIOME Study)' has been provisionally accepted for publication in PLOS Neglected Tropical Diseases.

Best regards,

Matthew Brian Rogers, Ph.D.

Academic Editor

Charles Jaffe

Section Editor

Thank-you for your the re-submission of you manuscript, after consideration of modifications made to your manuscript, your manuscript has been found suitable for publication. Please ensure that your deposited sequence data is released at the time of publication.

Reviewer's Responses to Questions

**Key Review Criteria Required for Acceptance?**

**Methods**

-Are the objectives of the study clearly articulated with a clear testable hypothesis stated?

-Is the study design appropriate to address the stated objectives?

-Is the population clearly described and appropriate for the hypothesis being tested?

-Is the sample size sufficient to ensure adequate power to address the hypothesis being tested?

-Were correct statistical analysis used to support conclusions?

-Are there concerns about ethical or regulatory requirements being met?

Reviewer #1: (No Response)

**Results**

-Does the analysis presented match the analysis plan?

-Are the results clearly and completely presented?

-Are the figures (Tables, Images) of sufficient quality for clarity?

Reviewer #1: (No Response)

**Conclusions**

-Are the conclusions supported by the data presented?

-Are the limitations of analysis clearly described?

-Do the authors discuss how these data can be helpful to advance our understanding of the topic under study?

-Is public health relevance addressed?

Reviewer #1: (No Response)

**Editorial and Data Presentation Modifications?**

Reviewer #1: (No Response)

**Summary and General Comments**

Reviewer #1: All my comments have been satisfactorily addressed.

PLOS authors have the option to publish the peer review history of their article (what does this mean?). If published, this will include your full peer review and any attached files.

Reviewer #1: No

---

## [Editor Report · Acceptance letter]

8 Jul 2023

Dear PhD Pérez-Molina,

We are delighted to inform you that your manuscript, "Chagas disease is related to structural changes of the gut microbiota in adults with chronic infection (TRIPOBIOME Study)," has been formally accepted for publication in PLOS Neglected Tropical Diseases.

Best regards,

Shaden Kamhawi

co-Editor-in-Chief

Paul Brindley

co-Editor-in-Chief
